# Copper(II) Complexes with 4-Substituted 2,6-Bis(thiazol-2-yl)pyridines—An Overview of Structural–Optical Relationships

**DOI:** 10.3390/ijms262411868

**Published:** 2025-12-09

**Authors:** Anna Maria Maroń, Anna Świtlicka, Agata Szłapa-Kula, Katarzyna Choroba, Karol Erfurt, Mariola Siwy, Barbara Machura

**Affiliations:** 1Institute of Chemistry, University of Silesia, Szkolna 9, 40-006 Katowice, Poland; agata.szlapa-kula@us.edu.pl (A.S.-K.); katarzyna.choroba@us.edu.pl (K.C.); barbara.machura@us.edu.pl (B.M.); 2Department of Chemical Organic Technology and Petrochemistry, Silesian University of Technology, Krzywoustego 4, 44-100 Gliwice, Poland; karol.erfurt@polsl.pl; 3Centre of Polymer and Carbon Materials, Polish Academy of Sciences, M. Curie-Skłodowska 34, 41-819 Zabrze, Poland

**Keywords:** copper(II), 2,2′:6′,2″-terpyridine derivatives, 2,6-bis(thiazol-2-yl)pyridine, X-ray structural analysis, optical properties, UV-Vis absorption spectroscopy

## Abstract

Copper(II) complexes with 2,2′:6′,2″-terpyridines (**terpys**) are promising candidates for anticancer therapy and catalysis. Their structural and optical properties can be tuned by modifying the **terpy** backbone, including a substitution at the 4′ position or the replacement of peripheral pyridines with thiazole rings, forming 2,6-bis(thiazol-2-yl)pyridines (**dtpys**). **dtpy**-based copper(II) complexes (**Cu-dtpys**), despite their applicative potential, are barely characterized in the literature. Here, the series of **Cu-dtpys** (**1**–**13**) was synthesised and characterized by FT-IR, HRMS, X-ray diffraction, and UV-Vis spectroscopy. Their structural and optical features were compared to previously studied **Cu-dtpys** (**14**–**24**) and their terpy analogues (**Cu-terpy-1** ÷ **Cu-terpy-24**). The detailed analysis revealed that five-coordinate **Cu-dtpys** complexes adopt a square pyramidal geometry comparable to that of **Cu-terpys** complexes but with markedly smaller deviations from the ideal square pyramid. Compared with **Cu-terpys**, Cu–Cl_apical_ bonds are shorter, while Cu–N_central_ bonds are elongated. The **Cu-dtpy** systems usually present the longest wavelength of the lowest energy absorption band in comparison to **Cuterpys**. The analysis of the relationship between Hammett’s constant and wavelength of absorption indicates that the most promising from the photophysical point of view are compounds **4**–**6**, **10**–**13**, **16**–**17**, and **22**, for which a newly formed intraligand charge transfer band is formed.

## 1. Introduction

Copper, after iron and zinc, is a key transition metal essential for the living body. It plays a role as a cofactor of many enzymes (e.g., cytochrome C oxidase, superoxide dismutase, tyrosinase) and is required for iron metabolism (in the form of ceruloplasmin), ATP production, and the formation of connective tissues and participates in the synthesis of neurotransmitters [1,2,3,4,5,6]. Moreover, copper-dependent enzymes are involved in white blood cell activity and are also responsible for oxygen stress reduction in living cells [3,7,8]. Five- and six-coordinate copper(II) complexes contribute significantly to anticancer drug development due to their oxidative, hydrolytic, and photolytic DNA-cleaving properties [9]. As an endogenous metal, the Cu(II) ion can create coordination compounds that are less toxic than platinum-based drugs and, owing to their selective permeability toward cancer cell membranes, can overcome drug transportation limitations [10]. Finally, copper(II) complexes are widely used as eco-friendly and versatile catalysts in oxidation, cross-coupling, radical polymerization, and biomimetic enzyme reactions [11,12,13].

Recently, our scientific attention was focused on five-coordinate copper(II) complexes with 2,2′:6′,2′′-terpyridine (**terpy**) derivatives as promising catalysts for oxidation of alkanes and alcohols with hydrogen peroxide and tert-butyl hydroperoxide (TBHP) and for use as antiproliferative drugs on A2780 ovarian and HCT116 human colorectal carcinoma cells [14,15,16]. **Terpy** and its derivatives, owing to their convenient synthesis via one-pot Kröhnke condensation, represent an ideal structural backbone for further modification, enabling systematic studies on the tuning of the biological, catalytic, and optical behavior of the resulting Cu(II) complexes.

One of the possible approaches for modulating the physicochemical properties of the resulting complexes is the substitution of peripheral pyridine rings of **terpy** with thiazole units, as in 2,6-bis(thiazol-2-yl)pyridine (**dtpy**). Compared to pyridines in the **terpy** framework, thiazole rings in **dtpy** differ in their σ-donor and π-acceptor abilities [17]. While terpyridyl Cu(II) coordination compounds (**Cu-terpy**) are among the most extensively studied systems, with successful applications in anticancer therapy and catalysis [18,19,20,21,22], Cu(II) complexes bearing **dtpy** derivatives (**Cu-dtpy**) remain exceedingly rare [14,15,16,23,24] (Figure 1). Notably, preliminary investigations of this class of compounds indicate their promising potential for both anticancer and catalytic applications [14,15,16,23,24].

In the current work, we designed a new series of [CuCl_2_(R-dtpy)] (**1**–**6**) and [CuCl_2_(R′-Ph-dtpy)] (**7**–**13**) to explore the impact of different types of substituents attached to the central pyridine ring of the **dtpy** core on the structure and optical properties of the [CuCl_2_(R-dtpy)] and [CuCl_2_(R′-Ph-dtpy)] systems (Figure 2). The R/R′ substituents introduced into the central pyridine or phenyl group differ in electron-donating or electron-withdrawing properties, and they also exhibit different steric effects. Furthermore, the structure–optical correlations of the newly synthesized Cu(II) complexes are discussed in comparison with those of the previously reported [CuCl_2_(R-dtpy)] (**14**–**24**) analogues (Figure 3) [14,15,16,23,24] and with the [CuCl_2_(R-terpy)] analogues (**Cu-terpy-1** ÷ **Cuterpy-24**) [14,15,16,25,26,27,28,29,30,31,32,33,34,35,36,37,38,39,40,41,42,43,44]. To the best of our knowledge, the current work represents the first comprehensive investigation of the role of substituents in determining the structural and optical properties of [CuCl_2_(R-dtpy)] and [CuCl_2_(R′-Ph-dtpy)] complexes.

## 2. Results

### 2.1. Synthesis and General Characterization

The syntheses of 2,6-di(thiazol-2-yl)pyridine (**L^1^**) and its substituted derivatives (**L^2^**–**L^13^**) were performed using the conventional one-pot Krönke method, which is based on the condensation of 2-acetylthiazole with the corresponding aldehyde and involves in situ pyridine ring closure in the presence of ammonia as a nitrogen donor source [45].

The copper(II) complexes (**1**–**13**) were isolated as green (**1**–**4**, **6**–**9**) or greenish red (**5**, **10**–**13**) solids from the reactions of **L^1^**–**L^13^** with CuCl_2_ in CH_3_OH–CHCl_3_ solution. The molecular formulae of **1**–**13** were determined by elemental analysis, high-resolution mass spectrometry (HR-MS) (Appendix A) and single crystal X-ray diffraction analysis (**1**–**11**). The mass spectra of all Cu(II) complexes were recorded in positive mode, and the base peak in the HR-MS spectra of **1**–**13** corresponded to [M−Cl]^+^, indicating high lability of one of the coordinated chloride ions, as expected for these systems [14,15].

FT-IR spectra of **1**–**13** confirmed the presence of the **dtpy**-based ligand in the coordination sphere. The characteristic peaks of R-**dtpy**/R′-Ph-**dtpy** appear in the ranges 3130–2850 cm^–1^ (aromatic C–H stretching vibrations), 1640–1510 cm^–1^ (ν_(C=N)_ and ν_(C=C)_ stretches), 1500–1000 cm^–1^(ν_(C–N)_ and ν_(C–C)_ vibrations) and 850–600 cm^–1^ (ν_(C–S)_ vibrations and aromatic C–H deformation vibrations). Stretching modes ν_(C≡N)_ of the ligand in **7** and **10** are observed at 2228 cm^–1^ and 2247 cm^–1^, respectively. The absorptions in the higher-energy region of the infrared spectra of the Cu(II) complexes (3500–3200 cm^–1^) are attributed to the ν_(O–H)_ stretching vibrations of lattice solvent molecules (Appendix A).

Compared with the ν_(C=N)_ and ν_(C=C)_ stretching vibrations of the free ligands, the corresponding bands in the FT-IR spectra of the [CuCl_2_(R-dtpy)] and [CuCl_2_(R′-Ph-dtpy)] complexes appear at higher frequencies. A similar trend is observed for the [CuCl_2_(R-terpy)] and [CuCl_2_(R′-Ph-terpy)] systems. However, in the case of the Cu(II) complexes with **dtpy** derivatives, the shifts in the ν_(C=N)_ and ν_(C=C)_ stretching vibrations relative to those of the free ligands are slightly smaller [14,15,16,23,24,25,26,27,28,29,46,47,48,49,50,51] (Appendix A).

### 2.2. X-Ray Analysis

The crystallographic details (Appendix A), selected bond distances and angles (Appendix A), and a summary of the intermolecular contacts detected in the structures of **1**–**11** (Appendix A) are provided in Appendix A. Perspective views of the asymmetric units of the structures of [CuCl_2_(R-dtpy)] (**1**–**6**) and [CuCl_2_(R′-Ph-dtpy)] (**7**–**11**), along with the atom numbering are given in Figure 1 and Figure 2, respectively.

X-ray analysis confirmed the formulation of the examined complexes as pentacoordinated [CuCl_2_L^n^], with the metal atom coordinated to three nitrogen donors of the 2,6-di(thiazol-2-yl)pyridine ligand and the two chlorine ions. The coordination geometry around the Cu(II) ion in all examined complexes is identical and can be best described as distorted square pyramidal, as supported by angular structural index parameter τ [52] and shape-measure *S*_Q_(P) parameters calculated with the SHAPE program based on Continuous Shape Measures (CShM) concept [53] (Table 1). For an ideal square pyramid (SPY) and trigonal bipyramid (TBY), the values of τ are 0 and 1, respectively. As far as *S*_Q_(P) parameters are concerned, the less distorted from the reference ideal shape, the smaller the *S*_Q_(P) value [54,55]. As shown in Table 1, a slightly greater deviation from the ideal square-pyramidal (SPY) geometry is observed for complexes **7** and **11**, whereas the remaining complexes exhibit τ values below 0.125, with their S_Q_(SPY) parameters being significantly lower than the corresponding S_Q_(TBY) values.The angular distortion of the coordination sphere of the Cu(II) ions in examined complexes from an ideal square-pyramid can be attributed to the κ^3^N-coordination of the R-dtpy ligand and the formation of two fused five-member chelate rings upon coordination. As a result, the bite angles N–Cu–N are noticeably smaller than ideal value 90° [77.2(2)–78.90(12)°] (Appendix A). The **dtpy** framework is approximately planar, with the dihedral angles between the mean planes of the central pyridine and terminal aromatic rings ranging from 0.24° to 8.35° (Table 2). More noticeable differences among the examined structures concern the twisting of the pendant subsistent relative to the central pyridine ring of the **dtpy** ligand. The largest dihedral angle between the central pyridine and the appended group, 32.38°, was observed for **7**, whereas the pendant substituent in complex **5** remains nearly coplanar with the central pyridine plane. In all examined complexes, the metal ion is displaced above the basal plane {Cl(1),N(1),N(2),N(3)} toward Cl(2) ranging from 0.269 Å in complex **5** to 0.465 Å in complex **7** (Table 2).$$S_{Q}(P)=\min\left[\frac{\sum_{i=1}^{n}|{\vec{q}}_{i}-{\vec{p}}_{i}{|}^{2}}{\sum_{i=1}^{n}|{\vec{q}}_{i}-{\vec{q}}_{0}{|}^{2}}\right]\times 100$$

Minor differences are observed in the bond distances and angles around Cu(II) ion in the examined complexes, depending on the type of substituent (Table 2 and Appendix A). As is typical for a square-pyramidal configuration, the apical Cu–Cl bond [2.4003(11)–2.5750(12) Å] is significantly longer than the basal Cu–Cl bond [2.2136(9)–2.265(2)Å], consistent with Jahn Teller distortion [31]. By analogy with related Cu(II) complexes containing **terpy**-like ligands [14,15,29,56], the Cu–N_central_ bond lengths [1.952(5)–1.991(3) Å] are noticeably shorter than those to the peripheral thiazolyl rings [2.032(6)–2.079(2) Å]. Based on the provided values, the largest variations in bond lengths as a function of the substituent are observed for the apical Cu–Cl distances. A noticeably elongated apical Cu–Cl bond is observed in complexes **4**, **5**, **8** and **9** (Table 2 and Appendix A).

Regarding the packing arrangement, the crystal structures of complexes **2**, **3**, **6** and **9** are composed solely of mononuclear [CuCl_2_L^n^] units. Crystal packing analysis (Mercury 4.0) [57] indicates that the dominant forces driving the self-aggregation of [CuCl_2_L^n^] molecules in the crystal lattice are weak C–H•••Cl hydrogen bonds (Appendix A) and π•••π interactions (Appendix A). The structures of the other compounds consist of [CuCl_2_L^n^] units and co-crystallized solvent molecules. However, solvent molecules could be satisfactorily modeled only for complexes 1, 4, and 5. In the case of **7**, **8**, **10** and **11**, solvent molecules were removed from the electron density map using the OLEX2 solvent mask command [58]. Water molecules (**1**, **4** and **5**) and methanol molecules (**4**) participate in the formation of hydrogen bonds, as shown in Figure 3. Additionally, the crystal packing of compound **1**, **4** and **5** is stabilized by non-classical intermolecular C–H•••Cl/C–H•••O and π•••π interactions. More pronounced differences in the Addison parameters are observed for complexes **7**, **8**, **17**, **22** and their corresponding terpyridyl analogues. The terpy analogues of **8**, **17**, and **22** are characterized by markedly higher τ values compared to the corresponding [CuCl_2_(R-dtpy)] complexes. Among the terpyridyl Cu(II) complexes, those exhibiting the highest Addison parameters include [CuCl_2_(4-Mepyrrterpy)] (τ = 0.56) and [CuCl_2_(4-MeOnaphtterpy)] (τ = 0.32) [14,27].

To gain deeper insight into the structural features of the investigated compounds and to elucidate the influence of the R/R′ substituents and peripheral rings, the structural data of compounds **1**–**11** were compared with previously reported [CuCl_2_(R-dtpy)] (**14**–**23**) and their [CuCl_2_(R-terpy)] analogues (Appendix A, Appendix A) [14,15,16,23,24]. A comparative analysis of the Addison (τ) and *S*_Q_(P) parameters for the dtpy- and terpy-based five-coordinate Cu(II) complexes (Figure 4) suggests that the replacing the peripheral pyridine rings of **terpy** with thiazole units in **dtpy** generally does not induce significant changes in either the Addison (τ) or *S*_Q_(P) parameters. Both **dtpy**- and **terpy**-based Cu(II) complexes predominantly adopt a square pyramidal geometry, although the number of Cu(II) complexes with τ > 0.20 is higher among the terpyridyl systems. More pronounced differences in the Addison parameters are observed for compounds **7**, **8**, **17**, **22** and their corresponding terpyridyl analogues. The **terpy** analogues of **8**, **17**, **22** are characterized by markedly higher values of τ values compared to the corresponding [CuCl_2_(R-dtpy)] complexes. Among the terpyridyl Cu(II) complexes, those exhibiting the highest Addison parameters include [CuCl_2_(4-Mepyrrterpy)] (τ = 0.56), [CuCl_2_(4-MeOnaphtterpy)] (τ = 0.32) [14,27].

With respect to bond lengths, in the vast majority of the analyzed complexes, replacing **terpy** with **dtpy** results in a shortening of the apical Cu–Cl bond (Figure 5 and Appendix A). Among the terpyridyl complexes, the longest apical Cu–Cl distances are observed for [CuCl_2_(1-naphtterpy)]∙2CH_3_OH (SUCNOX) [2.642(2) Å] [16], [CuCl_2_(4-furanterpy] (FECSAK) [2.6009(9) Å] [14] and [CuCl_2_(6-MeOnaphtterpy)] (SUCNUD) [2.6064(8) Å] [16]. In contrast, the bond lengths between the Cu(II) cation and nitrogen of the central pyridyl ring of **dtpy** framework in **Cu-dtpys**, varying between 1.96 Å and 2.00 Å, are markedly longer than the Cu-N_pyridyl_ bond lengths in **Cu-terpys** (Cu-N_pyridyl_ bond lengths in range 1.93–1.96 Å in **Cu-terpys**) (see Figure 6 and Appendix A).

Noticeable differences between the **Cu**–**terpy** and **Cu**–**dtpy** series are also evident when the bite and capture angles are considered. In general, the **Cu**–**terpy** complexes exhibit larger bite angles (78.4–80.0°) than those observed for the **Cu**–**dtpy** analogues (77.9–78.9°) (see Appendix A). Consequently, the capture angles follow the same structural relationship between the two families of complexes, with values of 155.4–158.8° for the **Cu**–**terpy** series and 152.7–156.2° for the **Cu**–**dtpy** counterparts (Figure 7).

### 2.3. Optical Properties

The optical properties of complexes **1**–**13** were investigated in methanolic solutions (c = 10 μM and c = 1 mM) and compared with those of complexes **14**–**22** and their terpyridine Cu(II) analogues reported in the literature [14,15,16,25,27]. Consistent with our previous findings [16], the **Cu-dtpy** systems exhibit good stability in alcoholic solutions (ethanol or methanol), in contrast to DMSO and water medium. The UV-Vis absorption spectra of the investigated compounds are shown in Figure 8 (see also Appendix A). Additionally, the electronic absorption maxima of **Cu-dtpys** and **Cu-terpys** are summarized in Appendix A. A comparison of the absorption spectra of the studied **Cu-dtpy** compounds (**1**–**13**) with those of the corresponding organic ligands (**L^1^**–**L^13^**) is shown in Appendix A. Notably, compounds **4**–**6** and **10**–**14** exhibit the most pronounced spectral changes relative to their corresponding ligands, reflected in the appearance of a new absorption band in the visible region. The absorption onsets of these systems shift by more than 3800 cm^−1^ compared to the free ligand, resulting in visible-range absorptions with extinction coefficients of 33,000–110,000 M^−1^ cm^−1^ for **4**–**6** and 12,000–19,000 M^−1^ cm^−1^ for **10**–**14.** For complexes **1**–**3** and **7**–**9**, the observed bathochromic shifts of the lowest in energy absorption relative to the free ligands are considerably smaller, ~2000 cm^−1^.

The substituent effect in compounds **1**–**13** and **14**–**22**, along with their corresponding **Cu-terpy** analogues, was analyzed using Hammett’s substituent constant σ_p_ values, which were calculated using Equation (1) and the Calculation of Hammett Sigma Constants v2024.02 tool [59].σ_p_ = 2.0849 + 0.2074 q_1_ + 28.4679 q_m_ + 28.9006 q_p_(1)

q_1_—the charge on the atom to which the substituent is directly attached;

q_m_, q_p_—the charges on the carbons in the *meta* and *para* positions of the phenyl ring.

The calculated values of σ_p_ were then correlated with the longest-wavelength absorption maxima of dilute solutions of the respective compounds (see also Figure 9 and Figure 10 and Appendix A). Compounds **1**–**22** generally follow the trend in which a decrease in the Hammett’s σ_p_ value is accompanied by a bathochromic shift of the absorption maximum (Figure 9). Significant deviations from this correlation are observed for **2**, **5**, **9**, **14**–**15**, and **20**–**22**. Notably, compared with their **Cu-terpy** analogues, the longest-wavelength absorption maxima of the [Cu(4-R-dtpy)] complexes are red-shifted (Appendix A). Correlating the capture angles with the maxima of the lowest-energy absorption in dilute solutions for both **Cu-terpy** and **Cu-dtpy** systems reveals that the **Cu-dtpys** complexes follow a trend in which a smaller capture angle, N_peripheral_-Cu-N_peripheral_, is associated with a higher-wavelength absorption (see Figure 11).

The UV-Vis spectrum of complex **1** in dilute methanolic solution shows three bands in the UV-region, all of which are intraligand in nature (π→π* and n→π*). The lowest-energy band displays a vibronic structure with a maximum at 344 nm and two shoulders at 361 nm and 328 nm. The introduction of the phenyl (compound **2**) and naphthyl substituents (compounds **14** and **15**) into the **dtpy** framework results in a progressive decrease in the onset absorption energy of the corresponding [Cu(4-R-dtpy)Cl_2_] complexes, from 26,455 cm^−1^ (in **1**) through 26,042 cm^−1^ (in **2**) until 25,253 cm^−1^ (in **14**) or 25,445 cm^−1^ (in **15**) (see Figure 8 and Appendix A). This shift is accompanied by hyperchromism and broadening of the spectrum [16]. The spectra of compounds **18**–**19** are similar in shape to those of **14** and **15** [15,16]. In contrast, the compounds **16**–**17**, bearing electron-donating methoxynaphthyl substituents, exhibit an additional absorption band in the range of 375–475 nm, which is likely attributable to an intraligand charge transfer (ILCT) transition [16].

For N/O/S-heterocyclic-substituted compounds **3**–**6**, the lowest-energy absorption band shows a progressive red shift in the order **3** → **4** → **6** → **5** relative to compounds **1**–**2**. When comparing compounds **4** and **5** to compound **21,** and compound **6** to compound **20,** the absorption bands are bathochromically shifted by 2093 cm^−1^ for **4**, 4880 cm^−1^ for **5**, and 2702 cm^−1^ for **6** [14]. In fact, the spectral profiles of **20** and **21** more closely resemble that of **3** than those of **4**–**6**. The latter three compounds (**4**–**6**) show greater similarity to **22**, which contains an electron-donating substituent and exhibits the formation of an additional ILCT band [14,15,16] (Appendix A). In particular, compound **5** with the bithiophene substituent deviates significantly from the correlation between the calculated Hammett’s constant and the absorption wavelength. The calculated σ_p_ value for the bithiophene group indicates slightly electron-withdrawing properties, whereas the strong bathochromic shift observed experimentally suggests the opposite behavior. This apparent discrepancy most likely arises from two competing effects. The inductive effect, resulting from the electronegative sulfur atoms, appears to dominate in theoretical calculations, leading to the classification of bithiophene as an electron-accepting unit. In contrast, the resonance effect arising from the π-electron delocalization predominates experimentally and identifies bithiophene as a strong electron-donating substituent.

In the series of **7**–**13**, compounds **7** and **8** exhibit UV-Vis spectra comparable to those of **1** and **2**, respectively. For compound **9**, which bears the biphenyl substituent, both hyperchromic and bathochromic effects are observed, although its spectral profile remains similar in shape to that of **1**. Complexes **10**–**13**, containing electron-donating substituents, display a new band in the 400–550 nm range, which can be attributed to an ILCT transition originating from charge delocalization from the electron-rich substituent to dtpy acceptor moiety. Overall, complexes [CuCl_2_(R′-Ph-dtpy)] show a good correlation between the Hammett’s constants (σ_p_) and the wavelength of the lowest-energy absorption (Figure 10).

In the UV-Vis spectra of concentrated samples (Figure 8 and Appendix A), all investigated **Cu-dtpy** complexes show a weak and very broad band in the 550–950 nm range, arising from overlapping d–d transitions (d_xy_→d_x_^2^_-y_^2^, d_yz_,d_xz_→d_x_^2^_-y_^2^ and d_z_^2^→d_x_^2^_-y_^2^) of the copper(II) ion. In comparison with terpirydyl Cu(II) systems, a pronounced red shift of this band is observed for the [Cu(R-dtpy)Cl_2_] compounds [14,15,16].

## 3. Materials and Methods

CuCl_2_·2H_2_O, 2-acetylthiazole, appropriate aldehydes and reagent-grade solvents for the synthesis were purchased from commercial suppliers (Sigma-Aldrich, Darmstadt, Germany; ABCR, St. Louis, MO, USA) and used as received. The 2,6-di(thiazol-2-yl)pyridine derivatives were obtained by condensation of 2-acetylthiazole with the corresponding aldehyde according to the procedure described in the literature, and their analytical data were in good agreement with those reported in refs. [23,45,46,60,61,62,63,64,65,66,67,68,69,70,71,72].

### 3.1. General Synthesis Procedure for [CuCl_2_L^n^]

CuCl_2_·2H_2_O (0.17 g, 1 mmol) dissolved in methanol (30 mL) was added dropwise to a chloroform solution of the corresponding ligands L^1^–L^13^ (1 mmol). The resulting solution was stirred at room temperature for 12 h and then left to stand for slow solvent evaporation. After several days, a green (**1**–**4**, **6**–**9**) or greenish-red (**5**, **10**–**13**) precipitate formed, which was collected by filtration, washed with small amounts of methanol and chloroform, and air-dried at room temperature. Crystals suitable for X-ray analysis were obtained by recrystallization from methanol or a methanol–chloroform mixture.

**[CuCl_2_L^1^]·H_2_O** (**1**·H_2_O): Yield 60%. HRMS (ESI): calcd for C_11_H_7_ClCuN_3_S_2_^+^ 342.9066; found 342.9068. Anal. calcd for C_11_H_7_Cl_2_N_3_S_2_Cu·H_2_O (397.77 g/mol): C 33.21, H 2.28 N 10.56%; found: 32.99, H 1.97, N 10.16%. IR (KBr, cm^−1^): 3425(m), 3121(w), 3099(s), 3068(s), 3009(w), 1812(w), 1613(m), 1598(s), 1563(m), 1497(s), 1477(s), 1460(s), 1396(w), 1371(s), 1329(w), 1302(w), 1254(s), 1195(s), 1156(m), 1087(w), 1066(m), 1029(w), 1014(w), 885(w), 855(w), 803(s), 786(s), 757(m), 732(m), 684(m), 638(s), 519(w). UV-Vis (MeOH, λ_max_/nm (ε/M^−1^ × cm^−1^)): 743 (78), 364sh (21,305), 344 (32,655), 304 (30,524), 274 (44,004) and 223 (30,893). Main crystallographic details: monoclinic, *P*2_1_/*n*, Z = 4, a = 8.1922(5) Å, b = 11.1975(6) Å, c = 15.9212(10) Å, β = 104.569(7)°, V = 1413.52(15) Å^3^, D_c_ = 1.869 g/cm^3^, F(000) = 796, R indices (all data): *R*_1_ = 0.0529 and *wR*_2_ = 0.0819.

**[CuCl_2_L^2^]·**½**H_2_O** (**2**·½ H_2_O): Yield 75%. HRMS (ESI): calcd for C_17_H_11_ClCuN_3_S_2_^+^ 418.9379; found 418.9379. Anal. calcd for C_17_H_13_Cl_2_N_3_S_2_OCu·½H_2_O (464.87 g/mol): C 43.92, H 2.60, N 9.04%; found: C 43.95, H 2.41, N 9.01%. IR (KBr, cm^−1^): 3426(w), 3116(w), 3070(s), 3017(w), 1610(s), 1552(m), 1496(w), 1483(s), 1463(w), 1434(s), 1354(w), 1325(w), 1252(m), 1198(s), 1151(w), 1081(m), 1013(w), 919(w), 871 (m), 818 (w), 787(s), 765(s), 736(m), 682(m), 654(m), 630(m), 608(w), 503(w), 449(w). UV-Vis (MeOH, λ_max_/nm (ε/M^−1^ × cm^−1^)): 742 (80), 366sh (21,260), 349 (30,520), 301 (75,960) and 203 (64,240). Main crystallographic details: monoclinic, *P*2_1_/*n*, Z = 8, a = 18.5235(10) Å, b = 8.4191(4) Å, c = 23.4437(12) Å, β = 102.266(5)°, V = 3572.6(3)Å^3^, D_c_ = 1.695 g/cm^3^, F(000) = 1832, R indices (all data): *R*_1_ = 0.0478 and *wR*_2_ = 0.0815.

**[CuCl_2_L^3^]** (**3**): Yield 70%. HRMS (ESI): calcd for C_16_H_10_ClCuN_4_S_2_^+^ 419.9331; found 419.9334. Anal. calcd for C_16_H_10_Cl_2_N_4_S_2_Cu (456.84 g/mol): C 42.07, H 2.21, N 12.26%; found: C 41.61, H 2.00, N 11.87%. IR (KBr, cm^−1^): 3425(m), 3129(m), 3110(w), 3093(m), 2996(m), 1612(s), 1584(m), 1557(m), 1494(m), 1486(m), 1470(s), 1430(s), 1354(m), 1324(m), 1311(w), 1253(s), 1224(m), 1191(s), 1169(w), 1077(m), 1061(w), 1020(m), 991(m), 930(w), 896(m), 829(w), 790(s), 754(m), 739(m), 666(m), 654(w), 642(w), 618(w), 608(w), 505(w). UV-Vis (MeOH, λ_max_/nm (ε/M^−1^ × cm^−1^)): 760 (100), 354 (84,895), 308 (68,145), 271 (63,030), 252 (66,145) and 207 (91,680). Main crystallographic details: monoclinic, *P*2_1_/*c*, Z = 4, a = 14.1745(9) Å, b = 11.9978(6) Å, c = 10.4487(6) Å, β = 107.157(7)°, V = 1697.87(19)Å^3^, D_c_ = 1.787 g/cm^3^, F(000) = 916, R indices (all data): *R*_1_ = 0.0374 and *wR*_2_ = 0.0692.

**[CuCl_2_L^4^]** (4): Yield 65%. HRMS (ESI): calcd for C_17_H_11_ClCuO_2_N_3_S_3_^+^ 482.8998; found 482.8998. Anal. calcd for C_17_H_11_Cl_2_N_3_S_2_O_2_Cu (519,92 g/mol): C 39.27, H 2.13, N 8.08%; found: C 38.89, H 2.50, N 8.24%. IR (KBr, cm^−1^): 3472(m), 3383(m), 3100(m), 3077(m), 2936(w), 1602(s), 1571(w), 1537(m), 1488(s), 1439(s), 1358(m), 1288(w), 1251(m), 1203(s), 1203(s), 1149(w), 1062(s), 1022(m), 949(m), 914(m), 860(w), 838(w), 788(m), 736(m), 691(w), 608(m). UV-Vis (MeOH, λ_max_/nm (ε/M^−1^ × cm^−1^)): 737 (83), 380 (33,499), 359 (36,645), 335 (44,282), 306 (52,248) and 275 (40,021). Main crystallographic details: triclinic, *P*$\bar{1}$, Z = 2, a = 7.5457(7) Å, b = 12.6550(10) Å, c = 13.2542(9) Å, α = 77.635(6)°, β = 86.523(6)°, γ = 80.957(7)°, V = 1220.45(17) Å^3^, D_c_ = 1.638 g/cm^3^, F(000) = 614, R indices (all data): *R*_1_ = 0.0748 and *wR*_2_ = 0.1140.

**[CuL^5^Cl_2_]·2H_2_O** (**5**·2H_2_O): Yield 65%. HRMS (ESI): calcd for C_19_H_11_ClCuN_3_S_4_^+^ 506.8820; found 506.8821. Anal. calcd for C_19_H_11_Cl_2_CuN_3_S_4_·2H_2_O (580,02 g/mol): C 39.34, H 2.61, N 7.24%; found: C 39.05, H 2.55, N 7.65%. IR (KBr, cm^−1^): 3495(s), 3101(w), 3068(w), 1606(vs), 1552(w), 1514(m), 1486(s), 1458(s), 1434(s), 1357(w), 1343(w), 1252(m), 1207(w), 1195(w), 1049(w), 1023(s), 855(w), 835(w), 821(w), 787(m), 760(w), 750(w), 728(m), 610(w), 527(w), 452(w). UV-Vis (MeOH, λ_max_/nm (ε/M^−1^ × cm^−1^)): 727 (88), 425 (54,509), 349 (48,544), 309 (57,361), 281 (57,918) and 208 (68,145). Main crystallographic details: monoclinic, *P*2_1_/*c*, Z = 4, a = 9.7299(4) Å, b = 15.0039(6) Å, c = 15.6869(9) Å, β = 98.481(5)° V = 2265.03(19)Å^3^, D_c_ = 1.701 g/cm^3^, F(000) = 1172, R indices (all data): *R*_1_ = 0.0764 and *wR*_2_ = 0.1261.

**[CuL^6^Cl_2_]** (**6**): Yield 60%. HRMS (ESI): calcd for C_17_H_11_ClCuON_3_S_3_^+^ 434.9328; found 434.9330. Anal. calcd for C_17_H_11_Cl_2_N_3_S_2_OCu (471.87 g/mol): 43.27, H 2.35, N 8.91%; found: 43.54, H 2.42, N 9.18%. IR (KBr, cm^−1^): 3489(m), 3040(m), 1637(w), 1598(vs), 1544(m), 1475(s), 1454(m), 1393(w), 1357(w), 1306(m), 1253(w), 1197(s), 1082(w), 1019(s), 973(m), 880(m), 786(m), 759(s), 638(w), 615(s), 591(s). UV-Vis (MeOH, λ_max_/nm (ε/M^−1^ × cm^−1^)): 746 (90), 389 (105,610), 306 (77,780), 276 (86,300) and 208 (86,565). Main crystallographic details: triclinic, *P*$\bar{1}$, Z = 2, a = 8.0883(5) Å, b = 10.1003(6) Å, c = 12.0986(7) Å, α = 65.536(5)°, β = 86.260(5)°, γ = 89.456(5)°, V = 897.57(10) Å^3^, D_c_ = 1.746 g/cm^3^, F(000) = 474, R indices (all data): *R*_1_ = 0.0743 and *wR*_2_ = 0.1245.

**[CuL^7^Cl_2_]·½H_2_O** (**7**·½H_2_O): Yield 75%. HRMS (ESI): calcd for C_18_H_10_ClCuN_4_S_2_^+^ 443.9331; found 443.9331. Anal. calcd for C_18_H_10_Cl_2_N_4_S_2_Cu·½H_2_O (489.89g/mol): C 44.13, H 2.26, N 11.44%; found: C 44.08, H 2.49, N 11.80%. IR (KBr, cm^−1^): 3471(s), 3090(s), 3072(s), 3048(s), 2228(s), 1607(s), 1547(m), 1493(m), 1483(s), 1449(s), 1405(m), 1353(m), 1253(m), 1197(s), 1098(w), 1081(w), 1020(w), 1011(w), 919(w), 896(m), 845(s), 830(m), 778(s), 753(s), 644(w), 612(w), 563(m), 548(w), 537(w), 497(w). UV-Vis (MeOH, λ_max_/nm (ε/M^−1^ × cm^−1^)): 742 (80), 352 (24,500), 292 (84,115) and 234 (44,375). Main crystallographic details: triclinic, *P*$\bar{1}$, Z = 2, a = 8.0631(5) Å, b = 11.0698(10) Å, c = 12.7644(10) Å, α = 70.692(8)°, β = 85.962(6)°, γ = 69.087(7)°, V = 1002.94(15) Å^3^, D_c_ = 1.592 g/cm^3^, F(000) = 482, R indices (all data): *R*_1_ = 0.0493 and *wR*_2_ = 0.1024.

**[CuL^8^Cl_2_]** (**8**): Yield 75%. HRMS (ESI): calcd for C_17_H_10_ClCuBrN_3_S_2_^+^ 496.8484; found 496.8490. Anal. calcd for C_17_H_10_BrCl_2_N_3_S_2_Cu (534.75 g/mol): C 38.18, H 1.88, N 7.86%; found:C 37.92, H 2.13, N 7.41%. IR (KBr, cm^−1^): 3507(s), 3437(s) 3103(w), 3088(s), 3070(w), 3031(w), 1608(s), 1588(m), 1495(w), 1481(s), 1451(m), 1397(m), 1354(w), 1254(m), 1241(w), 1204(s), 1154(w), 1124(w), 1077(m), 1024(w), 1003(m), 910(w), 831(s), 819(m), 784(m), 770(m), 655(w), 608(w), 531(m), 470(w). UV-Vis (MeOH, λ_max_/nm (ε/M^−1^ × cm^−1^)): 742 (83), 367sh (20,746), 351sh (31,592), 306 (74,253), 233 (40,373) and 207 (56524). Main crystallographic details: monoclinic, *P*2_1_/*c*, Z = 4, a = 13.3642(8) Å, b = 7.8994(6) Å, c = 22.1169(10) Å, β = 99.082(5)° V = 2305.6(2) Å^3^, D_c_ = 1.541 g/cm^3^, F(000) = 1052, R indices (all data): *R*_1_ = 0.0736 and *wR*_2_ = 0.1182.

**[CuL^9^Cl_2_]** (**9**): Yield 80%. HRMS (ESI): calcd for C_23_H_15_ClCuN_3_S_2_^+^ 494.9692; found 494.9697. Anal. calcd for C_23_H_15_Cl_2_N_3_S_2_Cu (531.94 g/mol): C 51.93, H 2.84, N 7.90%; found:C 52.09, H 2.87, N 8.34%. IR (KBr, cm^−1^): 3424(m), 3077(m), 2939(m), 1600(s), 1565(w), 1547(w), 1494(w), 1481(s), 1453(m), 1442(m), 1406(m), 1355(w), 1333(w), 1243(m), 1201(s), 1160(w), 1084(w), 1022(w), 1005(w), 877(w), 840(m), 828(w), 787(m), 766(s), 735(s), 693(m), 660(w), 610(m), 501(w). UV-Vis (MeOH, λ_max_/nm (ε/M^−1^ × cm^−1^)): 739 (100), 349 (89,440), 307 (97,925), 263 (76,290) and 208 (122,750). Main crystallographic details: triclinic, *P*$\bar{1}$, Z = 2, a = 8.0603(7) Å, b = 8.9973(7) Å, c = 15.1195(12) Å, α = 80.145(6)°, β = 80.145(6)°, γ = 89.634(7)°, V = 1074.34(15) Å^3^, D_c_ = 1.644 g/cm^3^, F(000) = 538, R indices (all data): *R*_1_ = 0.0889 and *wR*_2_ = 0.1431.

**[CuL^10^Cl_2_]** (**10**): Yield 65%. HRMS (ESI): calcd for C_21_H_17_ClCuN_5_S_2_^+^ 500.9910 found 500.9913. Anal. calcd for C_21_H_17_Cl_2_N_5_S_2_Cu (537.97 g/mol): C 46.89, H 3.19, N 13.02%. Found:C 47.28, H 3.13, N 13.43%. IR (KBr, cm^−1^): 3431(m), 3098(w), 3014(w), 2247(w), 1587(vs), 1533(m), 1484(s), 1455(w), 1410(w), 1384(w), 1361(w), 1336 (w), 1251(m), 1200(s), 1117(w), 1081(w), 1015(w), 960(w), 880(w), 822(w), 787(w), 766(w), 609(w), 532(w).UV-Vis (MeOH, λ_max_/nm (ε/M^−1^ × cm^−1^)):732 (65), 438 (12,840), 347 (13,410), 304 (19,040), 278 (18,135) and 209 (22,850). Main crystallographic details: triclinic, *P*$\bar{1}$, Z = 2, a = 7.7152(3) Å, b = 10.8824(5) Å, c = 14.8139(7) Å, α = 78.339(4)°, β = 84.515(3)°, γ = 82.548(3)°, V = 1204.72(9)Å^3^, D_c_ = 1.483 g/cm^3^, F(000) = 546, R indices (all data): *R*_1_ = 0.0489 and *wR*_2_ = 0.1052.

**[CuL^11^Cl_2_]**·**H_2_O** (**11**·H_2_O): Yield 60%. HRMS (ESI): calcd for C_19_H_16_ClCuN_4_S_2_^+^ 461.9801; found 461.9803. Anal. calcd for C_19_H_16_Cl_2_N_4_S_2_Cu·H_2_O (516.95g/mol): C 44.15, H 3.51, N 10.84%; found: C 44.50, H 3.75, N 10.45%. IR (KBr, cm^−1^): 3431(m), 3081(w), 2920(w), 2806(w), 1583(vs), 1539(m), 1484(s), 1440(w), 1410(w), 1377(m), 1337(w), 1252(m), 1212(m), 1171(m), 1081(w), 1012(m), 946(w), 820(w), 785(w), 754(w), 732(w), 608(w), 570(w), 513(w). UV-Vis (MeOH, λ_max_/nm (ε/M^−1^ × cm^−1^)): 724 (95), 456 (15,565), 350 (14,420), 337 (15,050), 309 (23,035) and 280 (20,440). Main crystallographic details: triclinic, *P*$\bar{1}$, Z = 6, a = 10.9146(5) Å, b = 13.4961(8) Å, c = 24.4952(12) Å, α = 89.902(4)°, β = 84.135(4)°, γ = 76.982(5)°, V = 3496.3(3) Å^3^, D_c_ = 1.422 g/cm^3^, F(000) =1518, R indices (all data): *R*_1_ = 0.1392 and *wR*_2_ = 0.1996.

**[CuL^12^Cl_2_]**·**H_2_O** (**12**·H_2_O): Yield 65%. HRMS (ESI): calcd for C_21_H_18_ClCuN_4_S_2_^+^ 487.9957; found 487.9958. Anal. calcd for C_21_H_18_Cl_2_CuN_4_S_2_·H_2_O(542.99 g/mol): C 46.45, H 3.71, N 10.32%; found: C 46.80, H 3.87, N 9.97%. IR (KBr, cm^−1^): 3414(m), 3084(w), 2967(w), 2842(w), 1581(vs), 1537(m), 1481(s), 1449(m), 1400(s), 1335(m), 1249(s), 1209(s), 1082(w), 1010(m), 963(w), 866(w), 816(m), 785(w), 749(w), 641(m), 610(w), 511(w). UV-Vis (MeOH, λ_max_/nm (ε/M^−1^ × cm^−1^)): 734 (110), 465 (18,300), 383 (7,810), 355 (13,850), 309 (24,815) and 281 (20,500).

**[CuL^13^Cl_2_]·H_2_O** (**13**·H_2_O): Yield 60%. HRMS (ESI): calcd for C_21_H_18_ClCuON_4_S_2_^+^ 503.9907; found 503.9910. Anal. calcd for C_21_H_18_Cl_2_CuN_4_OS_2_·H_2_O(558.99 g/mol): C 45.12, H 3.61, N 10.02%; found: C 45.18, H 3.27, N 9.93%. IR (KBr, cm^−1^): 3413(m), 3089(w), 2939(w), 1587(vs), 1530(m), 1483(s), 1448(m), 1413(w), 1388(w), 1363(w), 1332(w), 1219(s), 1114(m), 1080(w), 1013(w), 927(m), 821(m), 788(m), 638(m), 522(w). UV-Vis (MeOH, λ_max_/nm (ε/M^−1^ × cm^−1^)): 734 (95), 427 (12,010), 340 (18,050), 304 (24,090) and 278 (22,770).

### 3.2. Instrumentation

High-resolution mass spectrometry analyses were performed on a Waters Xevo G2 Q-TOF mass spectrometer (Waters Corporation, Milford, MA, USA) equipped with an ESI source operating in positive-ion modes. Full-scan MS data were collected from 100 to 1000 Da in the positive-ion mode with a scan time of 0.5 s. To ensure accurate mass measurements, data were collected in centroid mode, and mass calibration was performed during acquisition using leucine enkephalin solution as an external reference (Lock-Spray™, Chattanooga, TN, USA), which generated reference ions at *m*/*z* 556.2771 Da ([M + H]+) in the positive ESI mode. The accurate mass and elemental composition for the molecular ion adducts were calculated using the MassLynx 4.2 software (Waters, Milford, MA, USA) incorporated with the instrument (Appendix A). IR spectra were recorded on a Nicolet iS5 spectrophotometer in the spectral range 4000–400cm^−1^ with the samples in form of KBr pellets (Appendix A). Electronic spectra were obtained on a Nicolet Evolution 220 in the range 250–1000 nm in methanol. The X-ray diffraction data were collected on an Oxford Diffraction four-circle diffractometer Gemini A Ultra with Atlas CCD detector, using graphite-monochromated Mo Kα radiation (λ = 0.71073 Å) at room temperature. Diffraction data collection, cell refinement, and data reduction were performed using the CrysAlis^Pro^ software (version 40) [73]. The structures were solved by direct methods using SHELXS and refined by full-matrix least-squares on *F*^2^ using SHELXL-2014 [74]. All non-hydrogen atoms were refined anisotropically, and hydrogen atoms were placed in calculated positions and refined using a riding model: *d*(C–H) = 0.93 Å, *U*_iso_(H) = 1.2 *U*_eq_(C) (for aromatic) and *d*(C–H) = 0.96 Å, *U*_iso_(H) = 1.5 *U*_eq_(C) (for methyl and water). The methyl groups were allowed to rotate about their local threefold axis. For compounds **7**, **8**, **10** and **11**, solvent molecules (CH_3_OH or H_2_O), which could not be modelled satisfactorily, were removed from the electron density map using the OLEX2 solvent mask command [58]. Details of the crystallographic data collection, structural determination, and refinement for compounds **1**–**11** are given in Appendix A.

## 4. Conclusions

In this study, a new series of **Cu-dtpy** complexes (**1**–**13**) was successfully synthesized and comprehensively characterized by FT-IR spectroscopy, high-resolution mass spectrometry (HRMS), single-crystal X-ray diffraction, and UV–Vis spectroscopy. Comparative analyses with previously reported **Cu-dtpy** analogues (**14**–**24**) and related terpyridine-based complexes (**Cu-terpy-1** ÷ **Cu-terpy-24**) revealed distinct structural and electronic trends. The five-coordinate **Cu-dtpy** complexes exhibit a stronger preference for square-pyramidal coordination geometry compared to their **Cu-terpy** counterparts. Within the series of **Cu-dtpy** compounds, only **7** and **11** display τ exceeding 0.20, while in the group of **Cu-terpy**, higher τ values are observed for **Cu-terpy-8**, **Cu-terpy-16**, **Cuterpy-17** and **Cu-terpy-22**. Regarding structural features, the studies also revealed that the Cu–Cl_apical_ bonds in **Cu-dtpy** complexes are notably shorter, whereas the Cu–N_central_ bonds are relatively elongated. Spectroscopically, the **Cu-dtpy** series exhibits red-shifted lowest-energy absorption bands relative to the **Cu-terpy** analogues. Furthermore, correlation analyses between Hammett’s substituent constants and absorption maxima indicate that complexes **4**–**6**, **10**–**13**, **16**–**17**, and **22** display the most favorable photophysical characteristics, attributed to the emergence of an additional intraligand charge-transfer (ILCT) transition. The **Cu-dtpy** compounds exhibiting strong visible ILCT absorption may be highly sensitive to environmental factors. Therefore, further studies on their responses to external stimuli, as well as their potential applications in sensing, biosensing, and catalysis (including photocatalysis and biocatalysis), could be of significant importance in the near future.

## Data Availability

Data are contained within the article and Appendix A. Crystallographic data for **1**–**11** were deposited with the Cambridge Crystallographic Data Center. CCDC Numbers 2500835–2500845 contain the supplementary crystallographic data for **1**–**11**. Authors will release the atomic coordinates upon article publication. Copies of this information may be obtained free of charge from the Director, CCDC, 12 Union Road, Cambridge CB2 1EZ, UK (Fax: +44-1223-336033; e-mail: deposit@ccdc.cam.ac.uk or www.ccdc.cam.ac.uk (accessed on 6 November 2025)).

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
