# Peer review of "Copper(II) Complexes with 4-Substituted 2,6-Bis(thiazol-2-yl)pyridines—An Overview of Structural–Optical Relationships"

_ijms, 2025, doi:10.3390/ijms262411868_

Round 1
Reviewer 1 Report
Comments and Suggestions for Authors
The article is of good experimental quality and can be published after some revisions.
The authors should pay attention to the following aspects:
1) The synthetic approach to obtaining the ligands and its complexes with the yields of the reaction products should be added to Scheme 1 (a similar scheme was presented in the authors' publication https://pubs.acs.org/doi/full/10.1021/acs.jmedchem.4c01806).
Also, in Scheme 1, it would be better to display the bond by which the R substituent attaches to the molecules as a wavy line (the real view looks good for most molecules, but not for amine derivatives (compounds 10-13))
2) I recommend comparing the copper atom capture angle of the terpyridine- and 2,6-bis(thiazol-2-yl)pyridine-based ligand systems. This would clearly demonstrate the influence of the used peripheral heterocyclic fragment (transition from a 6-membered to a 5-membered heterocycle) on the spectral properties of the systems.
3) Compound 4, according to Figure 3, contains one water molecule and two methanol molecules in its structure. However, it is presented in a solvate-free form in the experimental section. Other compounds should be checked and their descriptions should be brought into consistency.
4) The authors compare the UV spectra of the copper complexes with each other, but I also recommend comparing them with the UV spectra of the ligand systems used. This will clearly demonstrate the influence of CuCl2 on changes in the spectral properties of the organic molecule. (This information should be added to the Supplementary Information.)
I also recommend that the authors conduct quantum chemical investigation of the obtained compounds to reliably interpret the absorption bands in the UV spectra, as well as electrochemical studies (since the compounds under consideration are of interest in the field of redox catalysis). I believe that such comprehensive studies will further attract attention to these compounds.
Author Response
List of Changes and Authors’ Replies
We thank the Reviewer for valuable comments and suggestions which helped us to improve the manuscript. All the changes in the manuscript has been made with blue colour.
Reviewer 1
The article is of good experimental quality and can be published after some revisions.
The authors should pay attention to the following aspects:
1) The synthetic approach to obtaining the ligands and its complexes with the yields of the reaction products should be added to Scheme 1 (a similar scheme was presented in the authors' publication https://pubs.acs.org/doi/full/10.1021/acs.jmedchem.4c01806).
Also, in Scheme 1, it would be better to display the bond by which the R substituent attaches to the molecules as a wavy line (the real view looks good for most molecules, but not for amine derivatives (compounds 10-13))
A: The scheme with synthetic procedure was added to the Scheme 2, together with range of yields of reaction characteristic for complexes studied.
2) I recommend comparing the copper atom capture angle of the terpyridine- and 2,6-bis(thiazol-2-yl)pyridine-based ligand systems. This would clearly demonstrate the influence of the used peripheral heterocyclic fragment (transition from a 6-membered to a 5-membered heterocycle) on the spectral properties of the systems.
A: We thanks the Reviewer for the advice. The capture angles of Cu-terpy and Cu-dtpy systems have been carefully checked and are in the range of 155.18-158.15° for Cu-terpys and in the range of 153.0-156.17° for Cu-dtpys. The same correlation has been found in terms of the bite angles. The Figures 7 and S15 have been added to the main manuscript and to the ESI, respectively, accompanied by the following text in manuscript:
“Noticeable differences between the Cu–terpys and Cu–dtpys series are also evident when the bite and capture angles are considered. In general, the Cu–terpy complexes exhibit larger bite angles (78.4–80.0°) than those observed for the Cu–dtpy analogues (77.9–78.9°) (see Figure S15; ESI†). Consequently, the capture angles follow the same structural relationship between the two families of complexes, with values of 155.4–158.8° for the Cu–terpy series and 152.7–156.2° for the Cu–dtpy counterparts (Figure 7).”
Then, the relation between the capture angle and wavelength of absorption have been performed and the Figure 11 and text below have been added to the discussion:
“Correlating the capture angles with maxima of the lowest in energy absorption in diluted solutions for both Cu-terpy and Cu-dtpy systems reveals that Cu-dtpys complexes follow a trend in which a smaller capture angle, Nperipheral-Cu-Nperipheral, is associated with a higher wavelength of absorption (see Figure 11).”
3) Compound 4, according to Figure 3, contains one water molecule and two methanol molecules in its structure. However, it is presented in a solvate-free form in the experimental section. Other compounds should be checked and their descriptions should be brought into consistency.
A: Thank you for your advice, the compounds have been checked and their descriptions have been brought into consistency. Any differences between solvent molecules existing in SXRD structure and detected in elemental analysis (compound 2, 4, 8 and 10) are due to fact that a) microcrystalline samples were recrystallized in order to obtain monocrystals suitable for single X-ray analysis; and b) microcrystalline samples were dried before elemenental analysis.
4) The authors compare the UV spectra of the copper complexes with each other, but I also recommend comparing them with the UV spectra of the ligand systems used. This will clearly demonstrate the influence of CuCl2 on changes in the spectral properties of the organic molecule. (This information should be added to the Supplementary Information.)
A: Thank you for this valuable recommendation. The comparison between absorption spectra of coordination compounds 1 – 13 and the corresponding ligands L1 – L13 have been shown on Figure S18 in ESI. The following text have been added to the main:
“A comparison between the absorption spectra of the studied Cu-dtpy compounds (1–13) with those of the corresponding organic ligands (L1 – L13) is presented in Figure S17, ESI†. Notably, compounds 4–6 and 10–14 exhibit the most pronounced spectral changes relative to the corresponding ligands, reflected in the appearance of a new absorption band in the visible region. The absorption onsets of these systems shift by more than 3800 cm⁻¹ compared to the free ligand, yielding visible-range absorptions with extinction coefficients of 33,000–110,000 M⁻¹ cm⁻¹ for 4–6 and 12,000–19,000 M⁻¹ cm⁻¹ for 10–14. For complexes 1–3 and 7–9, the observed bathochromic shifts of the lowest in energy absorption in relation to the free ligands are considerable smaller, ~ 2000 cm⁻¹.”
5) I also recommend that the authors conduct quantum chemical investigation of the obtained compounds to reliably interpret the absorption bands in the UV spectra, as well as electrochemical studies (since the compounds under consideration are of interest in the field of redox catalysis). I believe that such comprehensive studies will further attract attention to these compounds.
A: We fully agree with the Reviewer that electrochemical measurements and quantum-chemical investigations would provide valuable additional insights into the properties of the Cu(II) complex family presented here. In the present manuscript, however, we have chosen to focus primarily on the structural–optical relationships of the Cu(II) complexes with 4-substituted dtpys. The electrochemical characterization of compounds 1–13 and 14–24, together with complementary DFT calculations, will be the subject of our future investigations.

Reviewer 2 Report
Comments and Suggestions for Authors
This paper describes the syntheses, characterization, crystal structures and spectroscopic properties of Cu(II) complexes with 2,6-bis(thiazol-2-yl)pyridine (dtpy) and its 4-substituted derivatives on pyridine ring. Along with the previously reported Cu(II)dtpys complexes, their structures and spectroscopic properties are discussed in comparison with the corresponding terpy complexes. This is a carefully done study, and the findings are interesting. Thus, this paper is worth publishing in International Journal of Molecular Science with minor revisions. Some additional comments are listed below.
1) page 4, line 103: 8 and 11 --> 7 and 10
2) page 9, line 200−201: The “[CuCl2(4-MeOnaphtterpy)] (τ = 0.32)” is written twice.
Authors should check and revise them.
3) page 11, line 212: (blue squares) --> (red squares)
4) page 13, line 263: hiperchromism --> hyperchromism
Author Response
List of Changes and Authors’ Replies
We thank the Reviewer for valuable comments and suggestions which helped us to improve the manuscript. All the changes in the manuscript has been made with blue colour.
Reviewer 2 wrote:
This paper describes the syntheses, characterization, crystal structures and spectroscopic properties of Cu(II) complexes with 2,6-bis(thiazol-2-yl)pyridine (dtpy) and its 4-substituted derivatives on pyridine ring. Along with the previously reported Cu(II)dtpys complexes, their structures and spectroscopic properties are discussed in comparison with the corresponding terpy complexes. This is a carefully done study, and the findings are interesting. Thus, this paper is worth publishing in International Journal of Molecular Science with minor revisions. Some additional comments are listed below.
1) page 4, line 103: 8 and 11 --> 7 and 10
2) page 9, line 200−201: The “[CuCl2(4-MeOnaphtterpy)] (τ = 0.32)” is written twice.
Authors should check and revise them.
3) page 11, line 212: (blue squares) --> (red squares)
4) page 13, line 263: hiperchromism --> hyperchromism
Author’s response: All corrections have been done.
